

# Probabilistically sampled and spectrally clustered plant species using phenotypic characteristics

Aditya A. Shastri[1], Kapil Ahuja[1], Milind B. Ratnaparkhe[2] and Yann Busnel[3]

[1] Math of Data Science & Simulation (MODSS) Lab, Indian Institute of Technology Indore, Indore, India
[2] ICAR-Indian Institute of Soybean Research, Indore, India
[3] Network Systems, Cybersecurity and Digital Law Department, Institut Mines-Telecom Atlantique, Rennes, France

## ABSTRACT

Phenotypic characteristics of a plant species refers to its physical properties as cataloged by plant biologists at different research centers around the world. Clustering species based upon their phenotypic characteristics is used to obtain diverse sets of parents that are useful in their breeding programs. The Hierarchical Clustering (HC) algorithm is the current standard in clustering of phenotypic data. This algorithm suffers from low accuracy and high computational complexity issues. To address the accuracy challenge, we propose the use of Spectral Clustering (SC) algorithm. To make the algorithm computationally cheap, we propose using sampling, specifically, Pivotal Sampling that is probability based. Since application of samplings to phenotypic data has not been explored much, for effective comparison, another sampling technique called Vector Quantization (VQ) is adapted for this data as well. VQ has recently generated promising results for genotypic data. The novelty of our SC with Pivotal Sampling algorithm is in constructing the crucial similarity matrix for the clustering algorithm and defining probabilities for the sampling technique. Although our algorithm can be applied to any plant species, we tested it on the phenotypic data obtained from about 2,400 Soybean species. SC with Pivotal Sampling achieves substantially more accuracy (in terms of Silhouette Values) than all the other proposed competitive clustering with sampling algorithms (*i.e.* SC with VQ, HC with Pivotal Sampling, and HC with VQ). The complexities of our SC with Pivotal Sampling algorithm and these three variants are almost the same because of the involved sampling. In addition to this, SC with Pivotal Sampling outperforms the standard HC algorithm in both accuracy and computational complexity. We experimentally show that we are up to 45% more accurate than HC in terms of clustering accuracy. The computational complexity of our algorithm is more than a magnitude less than that of HC.

Corresponding author
Kapil Ahuja, kahuja@iiti.ac.in

## INTRODUCTION

Genetic diversity has been an important foundation of plant breeding from the inception of agriculture since it helps develop new plants to meet the growing food demand globally. The breeding process is a complex combination of multiple stages (*Louwaars, 2018*). The first stage involves discovery of the native characteristics where the selection of diverse parent donors is of paramount importance (*Swarup et al., 2020*). One way plant genetic diversity can be studied is by using their phenotypic characteristics (physical characteristics). This kind of analysis can be relatively easily done because a sufficiently large amount of data is available from different geographical areas. In the phenotypic context, which is our first focus, a few characteristics that play an important role are Days to 50% Flowering, Days to Maturity, Plant Height, 100 Seed Weight, Seed Yield Per Plant, Number of Branches Per Plant, etc.

Cluster analysis is an important tool to describe and summarize the variation present between different plant species (*Painkra et al., 2018*). Thus, clustering can be used to obtain diverse parents which, as mentioned above, is of utmost importance. It is obvious that after clustering, the species present in the same cluster would have similar characteristics, while those present in different clusters would be diverse. Phenotypic data for the species of different plants (*e.g.*, Soybean, Wheat, Rice, Maize, etc.) usually have enough variation for accurate clustering. However, if this data is obtained for the species of the same plant, then clustering becomes challenging due to less variation in the data, which forms our second focus.

Hierarchical Clustering (HC) is a traditional and standard method that is currently being used by plant biologists for grouping of phenotypic data (*Painkra et al., 2018*; *Sharma et al., 2014*; *Kahraman, Onder & Ceyhan, 2014*). However, this method has a few disadvantages. First, it does not provide the level of accuracy required for clustering similar species (*Rokach, 2009*). Second, HC is based upon building a hierarchical cluster tree (also called dendrogram), which becomes cumbersome and impractical to visualize when the data is too large. The second most common clustering algorithm that is being currently used widely is Unweighted Pair Group Method using Arithmetic Mean (UPGMA). This algorithm is a variant of HC, and hence, has the same two disadvantages as discussed above.

To overcome these two disadvantages, in this paper, we propose the use of the Spectral Clustering (SC) algorithm. SC is mathematically sound and is known to give one of the most accurate clustering results among the existing clustering algorithms (*Luxburg, 2007*). For genotypic data, we have recently shown substantial accuracy improvements by using SC as well (*Shastri et al., 2019*). Furthermore, unlike HC, SC does not generate the intermediate hierarchical cluster tree. To the best of our knowledge, this algorithm has not been applied to phenotypic data in any of the previous works (see the Literature Review section below).

HC, as well as SC, both are computationally expensive. They require substantial computational time when clustering large amounts of data (*Luxburg, 2007*; *Mullner, 2013*). Hence, we use sampling to reduce this complexity. Probability-based sampling techniques
have recently gained a lot of attention because of their high accuracy at reduced cost (*Tille, 2006*). Among these, Pivotal Sampling is most commonly used, and hence, we apply it to phenotypic data (*Chauvet, 2012*). Like for SC, using Pivotal Sampling for phenotypic data is also new. Recently, Vector Quantization (VQ) has given promising results for genotypic data (*Shastri et al., 2019*). Hence, here we adapt VQ for phenotypic data as well. This also serves as a good standard against which we compare Pivotal Sampling.

To summarize, in this article, we develop a modified SC with Pivotal Sampling algorithm that is especially adapted for phenotypic data. The novelty of our work is in constructing the crucial similarity matrix for the clustering algorithm and defining the probabilities for the sampling technique. Although our algorithm can be applied to any plant species, we test it on around 2,400 Soybean species obtained from Indian Institute of Soybean Research, Indore, India (*Gireesh et al., 2015*). In the experiments, we perform four sets of comparisons. *First*, we show that use of Pivotal Sampling does not deteriorate the cluster quality. *Second*, our algorithm outperforms all the proposed competitive clustering algorithms with sampling in terms of the accuracy (*i.e.* modified SC with VQ, HC with Pivotal Sampling, and HC with VQ). The computational complexities of all these algorithms are similar because of the involved sampling. *Third*, our modified SC with Pivotal Sampling doubly outperforms HC, which as earlier, is a standard in the plant studies domain. In terms of the accuracy, we are up to 45% more accurate. In terms of complexity, our algorithm is more than a magnitude cheaper than HC. *Fourth* and *finally*, we demonstrate the superiority of our algorithm by comparing it with two previous works that are closest to ours.

The rest of this article is organized as follows. Section "Literature Review" provides a brief summary of the previous studies on phenotypic data. The standard algorithms for Pivotal Sampling and SC are discussed in Section "Sampling and Clustering Algorithms". Section "Implementing Pivotal Sampling and Modified Spectral Clustering for Phenotypic Data" describes the crucial adaptations done in Pivotal Sampling and SC for phenotypic data. The data description, validation metric, and the experimental set-up are presented in Section "Methodology". Section "Results and Discussion" gives the experimental results. Finally, conclusions and future work are provided in Section "Conclusions and Future Work".

## LITERATURE REVIEW

In this section, we present some relevant previous studies on phenotypic data and the novelty of our approach. Broadly, these studies can be classified into two categories. The first category consists of the works that identify relationships between the different phenotypic characteristics (for example, lower plant height may relate to lower plant yield or vice versa). These works are discussed in Section "First Category Previous Studies". The second category consists of the studies that identify the species with dissimilar phenotypic characteristics for the breeding program. These studies are discussed in Section "Second Category Previous Studies". Finally, we present a set of works that belong to both the categories in Section "Both Categories Previous Studies".

## First category previous studies

*Immanuel et al. (2011)* in 2011 measured nine characteristics of 21 Rice species. Grain Yield (GY) was kept as the primary characteristic, and its correlations with all others were obtained. It was observed that characteristics like Plant Height (PH), Days to 50% Flowering (DF), Number of Tillers per Plant (NTP), Filled Grains per Panicle (FGP) and Panicle Length (PL) were positively correlated with GY. The remaining characteristics were negatively correlated with GY.

*Divya et al. (2015)* in 2015 recorded 21 characteristics of two Rice species. The authors investigated the association between Infected Leaf Area (ILA), Blast Disease Susceptibility (BDS), Number of Tillers per Plant (NTP), Grain Yield (GY) and others. The authors concluded that, for example, (a) ILA had a significant positive correlation with leaf's BDS, (b) NTP exhibited the highest association with GY.

*Gireesh et al. (2015)* in 2015 analyzed eight characteristics of 3,443 Soybean species. The authors sampled the species using two methods, and correlations of all the characteristics with each other for both the samples were estimated. It was observed that, for example, Days to 50% Flowering (DF) was positively correlated with Days to Pod Initiation (DPI) in both the samples, while Number of Pods Per Plant (NPPP) showed a negative correlation with Nodes Per Plant (NPP).

*Huang et al. (2018)* in 2018 studied six characteristics of 206 Soybean species. These characteristics were correlated with the three types of leaves; elliptical leaves, lanceolate leaves and round leaves. The authors deduced that Soybean plants with lanceolate leaves had maximum average Plant Height (PH), Number of Pods per Plant (NPP), Number of Branches per Plant (NBP), and 100-Seed Weight (SW), while Soybean plants with other two types of leaves had lower values of these characteristics.

*Carpentieri-Pipolo, de Almeida Lopes & Degrassi (2019)* in 2019 investigated 45 phenotypic characteristics of a Soybean specie. The authors then studied the effect of 20 bacteria isolated from roots, leaves, and stems on these characteristics (*i.e.* whether the bacteria had positive or negative activity on (correlation with) the 45 characteristics). For example, *Enterobacter Ludwigii* (EL) bacteria, which is isolated from leaves, showed a positive correlation with 25 characteristics (*e.g.*, Plant Growth Promotion (PGP)) and a negative correlation with remaining 20 characteristics (*e.g.*, Phenylacetic Acid (PAC) assimilation). For better exposition, the above five studies are summarized in Table 1.

## Second category previous studies

*Sharma et al. (2014)* in 2014 performed clustering of 24 synthetic Wheat species. Cluster analysis was performed using HC, and the species were grouped into three clusters using the polymorphic Inter Simple Sequence Repeat (ISSR) markers. The authors argued that species belonging to different clusters were diverse in terms of heat tolerance, and could be used to develop better heat tolerant specie.

*Kahraman, Onder & Ceyhan (2014)* in 2014 analyzed the field performance of 35 Common Bean species by grouping them. The authors used HC, and the species were clustered into three groups based upon the matrix of relationship between the species.

**Table 1 Summary of first category previous studies.** Here, ⇒ represents positive correlation and ⇏ represents negative correlation.

| Studies | Plant | # of species | Inferred relationship |
|---|---|---|---|
| *Immanuel et al. (2011)* | Rice | 21 | PH, DF, NTP, FGP, PL ⇒ GY |
| *Divya et al. (2015)* | Rice | 2 | ILA ⇒ BDS |
| *Gireesh et al. (2015)* | Soybean | 3,443 | DF ⇒ DPI and NPPP ⇏ NPP |
| *Huang et al. (2018)* | Soybean | 206 | Lanceolate leaves ⇒ max avg PH, NPP, NBP and SW |
| *Carpentieri-Pipolo, de Almeida Lopes & Degrassi (2019)* | Soybean | 1 | EL ⇒ PGP and EL ⇏ PAC |

**Table 2 Summary of second category previous studies.**

| Studies | Plant | # of species | Clustering algorithm | # of clusters | Development of better species |
|---|---|---|---|---|---|
| *Sharma et al. (2014)* | Wheat | 24 | HC | 3 | Heat tolerant |
| *Kahraman, Onder & Ceyhan (2014)* | Common Bean | 35 | HC | 3 | Promising species for breeding |
| *Painkra et al. (2018)* | Soybean | 273 | HC | 7 | Improved characteristics |
| *Islam et al. (2020)* | Rice | 10 | HC | 3 | Higher plant yield |

[1] Heterosis refers to the phenomenon in which a hybrid plant exhibits superiority over its parents in terms of Plant Yield or any other characteristic.

The species belonging to different clusters were considered diverse, and were used to select promising species for breeding.

*Painkra et al. (2018)* in 2018 performed clustering of 273 Soybean species. Here, the authors used HC, and the species were grouped into seven clusters using Pearson Correlation Coefficient. According to the authors, the species belonging to the distant clusters were more diverse such that choosing them maximized heterosis[1] in cross-breeding.

*Islam et al. (2020)* in 2020 clustered ten Upland Rice species. Here, HC was used and the species were grouped into three clusters using a similarity coefficient between the species. The authors identified the two best species that could be used to obtain new species having higher plant yield. As earlier, here also, we summarize the above four studies in Table 2 below.

### Both categories previous studies

*Fried, Narayanan & Fallen (2018)* in 2018 analyzed 11 characteristics of 49 Soybean species. The authors determined correlations between the root characteristics and other phenotypic characteristics. For example, Shoot Dry Weight (SDW) and Chlorophyll Index (CI) were positively correlated with Total Root Length (TRL) and Total Root Surface Area (TRSA), while Plant Height (PH) was negatively correlated with TRSA and Average Root Diameter (ARD). In this work, Principal Component Analysis (PCA) biplot was used to separate the species into seven clusters. According to the authors, this research was critical for Soybean improvement programs since it helped select species with the improved root characteristics.

*Stansluos et al. (2019)* in 2019 analyzed 22 phenotypic characteristics for 11 Sweet Corn species. For example, the authors showed a positive and significant correlation of Yield of

**Table 3 Summary of both categories previous studies.**

| Studies | Plant | # of species | Inferred relationship | Clustering algorithm | # of clusters | Development of better species |
|---------|-------|--------------|----------------------|---------------------|---------------|------------------------------|
| *Fried, Narayanan & Fallen (2018)* | Soybean | 49 | SDW, CI ⇒ TRL, TRSA PH ⇏ TRSA, ARD | PCA | 7 | Improved root characteristics |
| *Stansluos et al. (2019)* | Sweet corn | 11 | YME ⇒ ED, NME YME ⇏ TKW | HC | 4 | Better morphological capabilities |

Marketable Ear (YME) with Ear Diameter (ED) and Number of Marketable Ear (NME), while a negative correlation between YME and Thousand Kernel Weight (TKW). Cluster analysis was performed using HC, and the corn species were grouped into four clusters using the Ward Linkage. The authors inferred substantial variation in morphological and agronomic capabilities of different species. Again, we summarize the above two studies in Table 3 below.

With the focus on the study of genetic diversity using phenotypic data, we have multiple novel contributions as below.

1. We focus on the second category above, and perform grouping of several thousand species as compared to a few hundred in the articles cited above. Note that from the first category, *Gireesh et al. (2015)* did work with about three thousand species, and we do compare one aspect of our work with this previous work (more on this in the point 2a below).

2. Clustering becomes computationally expensive when the size of the data is very large. Hence, sampling is required to make the underlying algorithm scalable. Thus, we perform clustering on the sampled data rather than the full one, which is not done in any of the articles above. We have two more innovations in this aspect as below.

   a) We use a probability-based sampling technique (Pivotal Sampling as mentioned earlier) that is highly accurate, and forms a completely new contribution.
   We demonstrate the superiority of our sampling by comparing it with the one done in *Gireesh et al. (2015)*. This comparison is discussed towards the end of the Results section. Please note that *Gireesh et al. (2015)* only performed sampling and did not cluster their data.

   b) HC, which is the most common clustering algorithm (and some other sporadically used algorithms like *k*-means and UPGMA), do not provide the level of accuracy needed. Again, as earlier, we develop a variant of the SC algorithm, which is considered highly accurate, especially for phenotypic data. Use of SC in this context is also completely new. We show the dominance of our clustering algorithm over the one proposed in the most recent past work by *Islam et al. (2020)* towards the end of the Results section. Again, please note that Islam et al. only performed clustering and did not sample their data.

## SAMPLING AND CLUSTERING ALGORITHMS

In this section, we briefly discuss the standard algorithms for Pivotal Sampling and SC in the two subsections below.

### Pivotal sampling

This is a well-developed sampling theory that handles complex data with unequal probabilities. The method is attractive because it can be easily implemented by a sequential procedure, *i.e.* by a single scan of the data (*Deville & Tille, 1998*). Thus, the complexity of this method is $\mathcal{O}(n)$, where $n$ is the population size. It is important to emphasize that the method is independent of the density of the data.

Consider a finite population $U$ of size $n$ with its each unit identified by a label $i = 1, 2,\ldots, n$. A sample $S$ is a subset of $U$ with its size, either being random ($N(S)$) or fixed ($N$). Obtaining the inclusion probabilities of all the units in the population, denoted by $\pi_i$ with $i = 1, 2,\ldots, n$, forms an important aspect of this unequal probability sampling technique.

The pivotal method is based on a principle of contests between units (*Tille, 2006*). At each step of the method, two units compete to get selected (or rejected). Consider unit $i$ with probability $\pi_i$ and unit $j$ with probability $\pi_j$, then we have the two cases as below.

1. **Selection step** ($\pi_i + \pi_j \geq 1$): Here, one of the units is selected, while the other one gets the residual probability $\pi_i + \pi_j - 1$ and competes with another unit at the next step. More precisely, if $(\pi_i, \pi_j)$ denotes the selection probabilities of the two units, then

$$(\pi_i, \pi_j) = \begin{cases} (1, \pi_i + \pi_j - 1) & \text{with probability } \dfrac{1 - \pi_j}{2 - \pi_i - \pi_j} \\ (\pi_i + \pi_j - 1, 1) & \text{with probability } \dfrac{1 - \pi_i}{2 - \pi_i - \pi_j} \end{cases} \quad (1)$$

2. **Rejection step** ($\pi_i + \pi_j < 1$): Here, one of the units is definitely rejected (*i.e.* not selected in the sample), while the other one gets the sum of the inclusion probabilities of both the units and competes with another unit at the next step. More precisely,

$$(\pi_i, \pi_j) = \begin{cases} (0, \pi_i + \pi_j) & \text{with probability } \dfrac{\pi_j}{\pi_i + \pi_j} \\ (\pi_i + \pi_j, 0) & \text{with probability } \dfrac{\pi_i}{\pi_i + \pi_j} \end{cases} \quad (2)$$

This step is repeated for all the units present in the population until we get the sample of size $N(S)$ or $N$. The worst-case occurs when we obtain the last sample (*i.e.* $N^{th}$ sample) in the last iteration.

### Spectral clustering

Clustering is one of the most widely used techniques for exploratory data analysis with applications ranging from statistics, computer science, and biology to social sciences and psychology etc. It is used to get a first impression of data by trying to identify groups having "similar behavior" among them. Compared to the traditional algorithms such as

$k$-means, SC has many fundamental advantages. Results obtained by SC are often more accurate than the traditional approaches. It is simple to execute and can be efficiently implemented by using the standard linear algebra methods. The algorithm consists of four steps as below (*Luxburg, 2007*).

1. The first step in the SC algorithm is the construction of a matrix called the similarity matrix. Building this matrix is the most important aspect of this algorithm; better its quality, better the clustering accuracy (*Luxburg, 2007*). This matrix captures the local neighborhood relationships between the data points *via* similarity graphs and is usually built in three ways. The first such graph is a $\varepsilon$-neighborhood graph, where all the vertices whose pairwise distances are smaller than $\varepsilon$ are connected. The second is a $k$-nearest neighborhood graph, where the goal is to connect vertex $v_i$ with vertex $v_j$ if $v_j$ is among the $k$-nearest neighbors of $v_i$. The third and the final is the fully connected graph, where each vertex is connected with all the other vertices. Similarities are obtained only between the connected vertices. Thus, similarity matrices obtained by the first two graphs are usually sparse, while the fully connected graph yields a dense matrix.

   Let the $n$ vertices of a similarity graph be represented numerically by vectors $a_1$, $a_2$,…, $a_n$, respectively. Here, each $a_i \in \mathbb{R}^m$ is a column vector for $i = 1,…, n$. Also, let $a_i^l$ and $a_j^l$ denote the $l^{th}$ elements of vectors $a_i$ and $a_j$, respectively, with $l = 1,…, m$. There exist many distance measures to build the similarity matrix (*Cha, 2007*). We describe some common ones below using the above introduced terminologies.

   a) **City block distance:** (*Cha, 2007*) It is the special case of the Minkowski distance

   $$d_{ij} = \sqrt[p]{\sum_{l=1}^{m} |a_i^l - a_j^l|^p} \tag{3}$$

   with $p = 1$.

   b) **Euclidean distance:** (*Cha, 2007*) It is the ordinary straight line distance between two points in the Euclidean space. It is again the special case of the Minkowski distance, where the value of $p$ is taken as 2. Thus, it is given by

   $$d_{ij} = \sqrt{\sum_{l=1}^{m} (a_i^l - a_j^l)^2}. \tag{4}$$

   c) **Squared Euclidean distance:** (*Cha, 2007*) It is the square of the Euclidean distance, and is given by

   $$d_{ij} = \sum_{l=1}^{m} (a_i^l - a_j^l)^2. \tag{5}$$

   d) **Cosine distance:** (*Cha, 2007*) It measures the cosine of the angle between two non-zero vectors, and is given by

$$d_{ij} = 1 - \frac{a_i \cdot a_j}{\|a_i\| \|a_j\|}, \tag{6}$$

where, $\|\cdot\|$ denotes the Euclidean norm of a vector.

e) **Correlation distance:** (*Szekely, Rizzo & Bakirov, 2007*) It captures the correlation between two non-zero vectors, and is given by

$$d_{ij} = 1 - \frac{(a_i - \bar{a}_i)^t (a_j - \bar{a}_j)}{\sqrt{(a_i - \bar{a}_i)^t (a_i - \bar{a}_i)}\sqrt{(a_j - \bar{a}_j)^t (a_j - \bar{a}_j)}}, \tag{7}$$

where, $\bar{a}_i$ and $\bar{a}_j$ are the means of $a_i$ and $a_j$ multiplied with a vector of ones, respectively, and $t$ signifies the transpose operation.

f) **Hamming distance:** (*Norouzi, Fleet & Salakhutdinov, 2012*) It measures the number of positions at which the corresponding values of two vectors are different, and is given by

$$d_{ij} = \frac{\#(a_i^l \neq a_j^l)}{n}, \tag{8}$$

g) **Jaccard distance:** (*Matlab Documentation, 2006*) It again measures the number of positions at which the corresponding values of two vectors are different excluding the positions where both the vectors have zero values, and is given by

$$d_{ij} = \frac{\#[(a_i^l \neq a_j^l) \cap ((a_i^l \neq 0) \cup (a_j^l \neq 0))]}{\#[(a_i^l \neq 0) \cup (a_j^l \neq 0)]}. \tag{9}$$

2. Next, a matrix called the Laplacian matrix is constructed. This matrix is either non-normalized or normalized. The non-normalized Laplacian matrix is defined as

$$L = D - W, \tag{10}$$

where $W$ is the similarity matrix and $D$ is a diagonal matrix whose elements are obtained by adding together the elements of all the columns for every row of $W$. Normalized Laplacian matrix is again of two types: the symmetric Laplacian ($L_{sym}$) and the random walk Laplacian ($L_{rw}$). Both these matrices are closely related to each other and are defined as

$$L_{sym} = D^{-1/2} L D^{-1/2} = I - D^{-1/2} W D^{-1/2}. \tag{11}$$

$$L_{rw} = D^{-1} L = I - D^{-1} W. \tag{12}$$

Henceforth, the non-normalized Laplacian matrix is referred to as the Type-1 Laplacian, $L_{sym}$ as the Type-2 Laplacian, and $L_{rw}$ as the Type-3 Laplacian. In the literature, it is suggested to use the normalized Laplacian matrix instead of the non-normalized one, and specifically the Type-3 Laplacian (*Luxburg, 2007*).

3. Once we have the Laplacian matrix, we obtain the first $k$ eigenvectors $u_1,\ldots, u_k$ of this matrix, where $k$ is the number of clusters.

4. Finally, these eigenvectors are clustered using the $k$-means clustering algorithm.

## IMPLEMENTING PIVOTAL SAMPLING AND MODIFIED SPECTRAL CLUSTERING FOR PHENOTYPIC DATA

Here, we first present the application of Pivotal Sampling to obtain the samples from phenotypic data. Subsequently, we implement our modified SC algorithm on the same data. Consider that the phenotypic data of a plant consists of $n$ species with each specie evaluated for $m$ different characteristics/traits. These characteristics may have categorical (non-numerical) or numerical values. Hence, we need to convert the categorical values into numerical ones. For this, we use the label encoder method (*Hancock & Khoshgoftaar, 2020*). This method transforms non-numerical labels into numerical values between 0 and (number of categories) – 1. For example, if a characteristic has three possible labels; poor, good, and very good, we use 0, 1, and 2 to represent them, respectively.

As discussed in Section "Pivotal Sampling", Pivotal Sampling requires that the inclusion probabilities (*i.e.* $\pi_i$ for $i = 1,\ldots, n$), of all the species in the population $U$, be computed before a unit is considered for a contest. The set of characteristics associated with a specie can be exploited in computing these probabilities. To select a sample of size $N$, where $N \ll n$, we obtain these probabilities as *Deville & Tille (1998)*

$$\pi_i = N \frac{\varkappa_i}{\sum_{i \in U} \varkappa_i},\tag{13}$$

where $\varkappa_i$ can be a property associated with any one characteristic (or a combination of them) of the $i^{th}$ specie. Obtaining $\pi_i$ in such a way also ensures that $\sum_{i=1}^{n} \pi_i = N$, *i.e.* we get exactly $N$ selection steps, and in-turn, exactly $N$ samples.

In our implementation, we use the deviation property of the species, which is discussed next. Since different characteristics have values in different ranges, we start by normalizing them as below (*Jain, Nandakumar & Ross, 2005*; *Shastri, Tamrakar & Ahuja, 2018*).

$$(\mathscr{X}_j)_i = \frac{(x_j)_i - \min(x_j)}{\max(x_j) - \min(x_j)}.\tag{14}$$

Here, $(\mathscr{X}_j)_i$ and $(x_j)_i$ are the normalized value and the actual value of the $j^{th}$ characteristic for the $i^{th}$ specie, respectively with $j = 1,\ldots, m$ and $i = 1,\ldots, n$. Furthermore, $\max(x_j)$ and $\min(x_j)$ are the maximum and the minimum values of the $j^{th}$ characteristic among all the species. Now, the deviation for the $i^{th}$ specie is calculated using the above normalized values as

$$dev_i = \sum_{j=1}^{m} \max(\mathscr{X}_j) - (\mathscr{X}_j)_i.\tag{15}$$

Here, $\max(\mathscr{X}_j)$ denotes the maximum normalized value of the $j^{th}$ characteristic among all the species. Practically, a relatively large value of $dev_i$ indicates that the $i^{th}$ specie is less

important, and hence, its probability should be small. Thus, the inclusion probability of a specie is calculated by taking $\varkappa_i = \frac{1}{dev_i}$ in Eq. (13) or

$$\pi_i = N \frac{\frac{1}{dev_i}}{\sum_{i \in U} \frac{1}{dev_i}} . \tag{16}$$

Thus, if the sum of probabilities of two species under consideration is greater than or equal to 1, we follow the selection step as discussed in Section "Pivotal Sampling". On the other hand, we follow rejection step when this sum is less than 1. This process is repeated until we obtain $N$ species.

Next, we discuss the clustering of these $N$ species into $k$ clusters. Similar to the standard SC algorithm discussed in Section "Spectral Clustering", the first step in our modified SC is to obtain the similarity matrix. As mentioned earlier, this is the most important aspect of this algorithm since the better the matrix quality, the better the clustering accuracy. For this, we consider these $N$ species as the vertices of a graph. Let vector $p_i$ contain the normalized values of all the characteristics ($m$) for the $i^{th}$ specie. Thus, we have $N$ such vectors corresponding to the $N$ species selected using Pivotal Sampling. That is, $p_i = [(\mathscr{X}_1)_i, ..., (\mathscr{X}_m)_i]^T$ for $i = 1,..., N$. In our implementation, we use a fully connected graph to build the similarity matrix, *i.e.* we obtain similarities among all the $N$ species.

We define the similarity between the vectors $p1$ and $p2$ (without loss of generality, representing the species 1 and 2, respectively) as the inverse of the distance between these vectors obtained by using the distance measures mentioned in "Spectral Clustering". This is intuitive because smaller the distance between any two species, larger the similarity between them and vice versa. We denote this distance by $d_{p1p2}$. We build this matrix of size $N \times N$ by obtaining the similarities among all the $N$ species.

The next step is to compute the Laplacian matrix, which when obtained from the above-discussed similarity matrix, generates poor eigenvalues,[2] and in-turn poor corresponding eigenvectors that are required for clustering[3]. Thus, instead of taking only the inverse of $d_{p1p2}$, we also take its exponent, *i.e.* we define the similarity between the species 1 and 2 as $e^{-d_{p1p2}}$ (*Ng, Jordan & Weiss, 2002*; *Nemade et al., 2018*). This, besides fixing the poor eigenvalues/eigenvectors problem, also helps perform better clustering of the given data. Further, we follow the remaining steps as discussed in "Spectral Clustering".

Above, we discussed the clustering of $N$ sampled species into $k$ clusters. However, our goal is to cluster all $n$ species and not just $N$. Hence, there is a need to reverse-map the remaining $n - N$ species to these $k$ clusters. For this, we define the notion of average similarity, which between the non-clustered specie $\tilde{p}$ and the cluster $C_l$ is given as

$$\mathscr{AS}(C_l, \tilde{p}) = \frac{1}{\#(C_l)} \sum_{q \in C_l} e^{-d_{\tilde{p}q}} . \tag{17}$$

Here, $\#(C_l)$ denotes the number of species present in $C_l$ and $q$ is a specie originally clustered in $C_l$ by our modified SC algorithm with Pivotal Sampling. We obtain the average

[2] Zero/close to zero and distinct eigenvalues are considered to be a good indicator of the connected components in a similarity matrix. Thus, eigenvalues are considered poor when they are not zero/not close to zero or indistinct (*Luxburg, 2007*).

[3] For some distance matrices (like Euclidean distance), the eigenvalues don't even converge.

similarity of $\tilde{p}$ with all the $k$ clusters (*i.e.* with $C_l$ for $l = 1,…, k$), and associate it with the cluster with which $\tilde{p}$ has the maximum similarity.

Next, we perform the complexity analysis of our algorithm. Since Pivotal Sampling and SC form the bases of our algorithm, we discuss the complexities of these algorithms before ours.

1. Pivotal Sampling ($n$: number of species, $N$: sample size)
(a) Obtaining Samples: $\mathcal{O}(n)$

2. SC ($n$, $m$: number of characteristics)
(a) Constructing Similarity Matrix: $\mathcal{O}(n^2 m)$
(b) Obtaining Laplacian Matrix: $\mathcal{O}(n^3)$

3. Our Algorithm ($n$, $N$, $m$)
(a) Obtaining Samples: $\mathcal{O}(n)$
(b) Constructing Similarity Matrix: $\mathcal{O}(N^2 m)$
(c) Obtaining Laplacian Matrix: $\mathcal{O}(N^3)$
(d) Reverse Mapping: $\mathcal{O}((n - N)N)$

Thus, the overall complexity of our algorithm is $\mathcal{O}(nN + N^3 + N^2 m)$. Here, we have kept three terms because any of these can dominate (here, $n \gg N, m$).

When we compare complexity of our algorithm with that of HC, which is $\mathcal{O}(n^3)$, it is evident that we are more than a magnitude faster than HC. We revisit this complexity analysis after discussing data in the next section, which supports our claim further.

## METHODOLOGY

In this section, we first briefly discuss the data used for our experiments. Next, we check the goodness of our sampling technique by estimating a measure called the population total. The hypothesis related to this is as follows: for a particular sampling technique, if the estimate (or approximation) of the population total using the samples is close to the actual population total, then that sampling technique is considered good in an absolute sense. Finally, we describe the clustering set-up, where the below are analyzed.

a) **The Validation Metric.** It is hypothesized that a good clustering is one where clusters are compact and well-separated.

b) **The Ideal Number of Clusters.** The hypothesis related to this is as follows: given a set of eigenvalues of the Laplacian matrix, we can exploit the differences between these eigenvalues to obtain the ideal number of clusters.

c) **The Suitable Distance Measures.** For building the similarity matrix and the Laplacian matrix, it is hypothesized to chose those matrices that give the best value for the validation metric.

**Table 4 Computational complexity comparison for the given data.**

| # of species ($n$) | # of characteristics ($m$) | Sample size ($N$) | Our algorithm ($nN + N^3 + N^2 m$) | HC ($n^3$) |
|---|---|---|---|---|
| 2,376 | 8 | 500 | $(2,376 \times 500) + (500)^3 + (500)^2 \times 8 = 1.28 \times 10^8$ | $(2,376)^3 = 1.34 \times 10^{10}$ |
| 2,376 | 8 | 300 | $(2,376 \times 300) + (300)^3 + (300)^2 \times 8 = 2.84 \times 10^7$ | $(2,376)^3 = 1.34 \times 10^{10}$ |

## Data description

As mentioned in Introduction, our techniques can be applied to any plant data, however, here we experiment on phenotypic data of Soybean species. This data is taken from Indian Institute of Soybean Research, Indore, India, and consists of 29 different characteristics/ traits for 2,376 Soybean species (*Gireesh et al., 2015*). Among these, we consider the following eight characteristics that are most important for higher yield: Early Plant Vigor (EPV), Plant Height (PH), Number of Primary Branches (NPB), Lodging Score (LS), Number of Pods Per Plant (NPPP), 100 Seed Weight (SW), Seed Yield Per Plant (SYPP) and Days to Pod Initiation (DPI). Out of these, EPV and LS have categorical values, while the remaining characteristics have numerical values. Hence, we convert these two categorical values into numerical ones using the label encoder method discussed in the previous section. A snapshot of this phenotypic data for a few Soybean species is given in Appendix A. Here, we also perform validation of this data by comparing it with a similar dataset.

Next, we compare the complexities of our algorithm and HC using the selected data; see Table 4. It is evident from this table that our algorithm achieves substantial savings.

## Sampling discussion

To inspect the quality of our sampling techniques, we estimate a measure called the population total, which is the addition of values of a particular characteristic for all the $n$ units (species here) present in the population $U$. For example, if "Plant Height (PH)" is the characteristic of interest, then the population total is the addition of PH values for all the $n$ species. Mathematically, the exact (or actual) population total for a characteristic of interest $x_j$ is given as

$$Y = \sum_{i \in U} (x_j)_i, \tag{18}$$

where, as earlier, $(x_j)_i$ is the value of the $j^{th}$ characteristic for the $i^{th}$ specie and $U$ is the set of all species. By the definition of this measure (and also for two more measures listed below in this section), we work with original (non-normalized) values of the characteristics rather than normalized ones. Also, based upon the same argument, we work with only those characteristics that are originally numerical.

In this work, we use two different estimators to compute an approximation of the population total from the sampled data. Closer the value of an estimator to the actual value, better the sampling. First is the Horvitz–Thompson (HT)-estimator (also called $\pi$-estimator), which is defined as *Horvitz & Thompson (1952)*

**Table 5 HT and Hájek estimator values for pivotal sampling and VQ as compared to the actual population total with $N = 500$ as the sample size.**

| Sr. No. | Characteristics | Actual population total | Pivotal sampling (HT) | VQ (HT) | Pivotal sampling (Hájek) | VQ (Hájek) |
|---|---|---|---|---|---|---|
| 1 | PH | 121,773.05 | 122,507.84 | 123,407.80 | 123,716.09 | 113,168.90 |
| 2 | NPB | 8,576.56 | 8,585.28 | 9,669.29 | 8,669.95 | 8,867.05 |
| 3 | NPPP | 99,712.72 | 100,193.53 | 114,465.66 | 101,181.70 | 104,968.67 |
| 4 | SW | 20,073.32 | 19,907.10 | 20,966.86 | 20,103.44 | 19,227.28 |
| 5 | SYPP | 10,048.04 | 10,137.57 | 10,536.08 | 10,237.55 | 9,661.92 |
| 6 | DPI | 136,810 | 135,309.78 | 149,242.17 | 136,644.29 | 136,859.84 |

$$Y'_{HT} = Y'_{\pi} = \sum_{i \in S} \frac{(x_j)_i}{\pi_i}, \tag{19}$$

where, $\pi_i$ is the inclusion probability of the $i^{th}$ specie as evaluated in Section "Implementing Pivotal Sampling and Modified Spectral Clustering for Phenotypic Data" and $S$ is the set of sampled species. Another estimator that we use is the Hájek-estimator. It is usually considered better than the HT-estimator and is given as *Hájek (1971)*

$$Y'_{Hájek} = n \frac{\sum_{i \in S} \frac{(x_j)_i}{\pi_i}}{\sum_{i \in S} \frac{1}{\pi_i}}, \tag{20}$$

here, as earlier, $n$ is the total number of species.

The actual population total and the values of the above two estimators for six characteristics (that have numerical values) when using Pivotal Sampling and 500 samples are given in Table 5 (see columns 3, 4, and 6, respectively). From this table, it is evident that the approximate values of the population total are very close to the corresponding actual values. Thus, Pivotal Sampling works well in an absolute sense. Here, we also compute the values of the two estimators when using VQ (see columns 5 and 7). We can notice from these results that VQ also works reasonably well, but Pivotal Sampling is better.

## Clustering setup

Here, *first*, we describe the criteria used to check the goodness of the generated clusters. There are two categories of metrics available for the validation of clustering algorithms. One category includes the metrics that require prior knowledge of the cluster labels (*Fahad et al., 2014*). On the other hand, metrics from the second category do not have this requirement (*Fahad et al., 2014; Rousseeuw, 1987*). In this work, the ideal cluster labels are not available, and hence, we use a metric called Silhouette Value (from the second category) for validation of our clustering algorithms (*Rousseeuw, 1987*).

Clustering is considered good if the obtained clusters are compact and well-separated. Silhouette Value captures both these aspects well by computing the intra-cluster similarity and the inter-cluster similarity. Consider that we have $k$ clusters represented

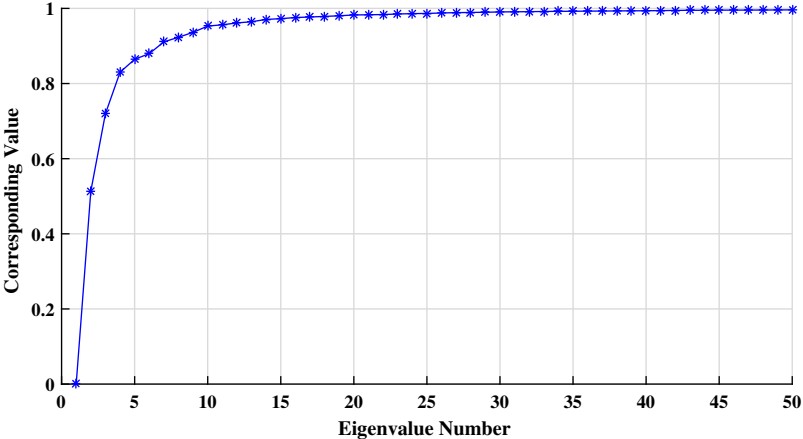

**Figure 1 Fifty smallest eigenvalues of the Type-3 Laplacian matrix obtained from the Euclidean similarity matrix (for estimating the ideal number of clusters).**

as $C_1,\ldots, C_k$, and we want to obtain the Silhouette Value of the $i^{th}$ data point present in the cluster $C_1$. For this, we compute the average distance between this data point and all the other points in the cluster $C_1$. This distance is denoted as $a(i)$. Next, we compute the average distance between the $i^{th}$ data point and all the other points in clusters $C_2,\ldots, C_k$. This distance is denoted as $b(i)$. Then, for this point, Silhouette Value is computed as below (*Rousseeuw, 1987*).

$$s(i) = \frac{b(i) - a(i)}{\max\{a(i), b(i)\}}. \tag{21}$$

As evident from the above discussion, the intra-cluster similarity is captured by $a(i)$, and the inter-cluster similarity is captured by $b(i)$. This value usually lies between minus one to plus one because the denominator of Eq. (21) is always greater than its numerator. Silhouette Value for the overall clustering is obtained by averaging the Silhouette Values of all the data points. If this value tends towards a positive one, then the clustering is considered to be good. On the other hand, if this value tends towards a negative one, then the clustering is considered poor.

*Second*, we determine the ideal number of clusters by using the eigenvalue gap heuristic (*Luxburg, 2007*; *Kong et al., 2010*). If $\lambda_1, \lambda_2, \ldots, \lambda_n$ are the eigenvalues of the matrix used for clustering (*e.g.*, the Laplacian matrix), then often the initial set of eigenvalues, say $k$, have a considerable difference between the consecutive ones in this set. That is, $|\lambda_i - \lambda_{i+1}| \not\approx 0$ for $i = 1,\ldots, k-1$. After the $k^{th}$ eigenvalue, this difference is usually approximately zero. According to this heuristic, this $k$ gives a good estimate of the ideal number of clusters.

For this experiment, without loss of generality, we build the similarity matrix using the Euclidean distance measure on the above discussed phenotypic data. As mentioned earlier, it is recommended to use the Type-3 Laplacian matrix (*Luxburg, 2007*). Hence, we use its eigenvalues for estimating $k$. Figure 1 represents the graph of the first fifty smallest

**Table 6 Silhouette values for modified SC with seven similarity measures and three Laplacian matrices for $k$ = 10, 20, and 30.** Silhouette values in bold represent good clustering.

| Sr. No. | Similarity measure | Number of clusters ($k$) | Type-1 Laplacian | Type-2 Laplacian | Type-3 Laplacian |
|---------|--------------------|--------------------------|------------------|------------------|------------------|
| 1. | Euclidean | 10 | 0.0828 | −0.0273 | **0.2422** |
|  |  | 20 | 0.0455 | −0.1096 | **0.2069** |
|  |  | 30 | 0.0887 | −0.1536 | **0.1783** |
| 2. | Squared Euclidean | 10 | 0.0815 | −0.0555 | **0.3836** |
|  |  | 20 | −0.0315 | −0.1809 | **0.2612** |
|  |  | 30 | 0.0354 | −0.2367 | **0.1538** |
| 3. | City-block | 10 | 0.0687 | 0.2375 | **0.2647** |
|  |  | 20 | −0.0356 | 0.1347 | **0.2082** |
|  |  | 30 | −0.0870 | 0.0866 | **0.1887** |
| 4. | Cosine | 10 | 0.1737 | −0.1408 | 0.0694 |
|  |  | 20 | 0.0359 | −0.1973 | 0.0277 |
|  |  | 30 | 0.0245 | −0.2456 | −0.0316 |
| 5. | Correlation | 10 | 0.1926 | −0.1259 | **0.3426** |
|  |  | 20 | 0.0970 | −0.2198 | **0.2313** |
|  |  | 30 | 0.2383 | −0.2604 | **0.1556** |
| 6. | Hamming | 10 | 0.0643 | 0.0706 | 0.0775 |
|  |  | 20 | 0.0683 | 0.0311 | 0.0382 |
|  |  | 30 | 0.0715 | 0.0283 | 0.0229 |
| 7. | Jaccard | 10 | 0.0716 | 0.0303 | 0.0458 |
|  |  | 20 | 0.0446 | 0.0276 | 0.0236 |
|  |  | 30 | 0.0279 | 0.0298 | 0.0318 |

eigenvalues (in absolute terms) of this Laplacian matrix. On the *x*-axis, we have the eigenvalue number, and on the *y*-axis its corresponding value.

From this figure, we can see that there is a considerable difference between the first ten consecutive eigenvalues. After the tenth eigenvalue, this difference is very small (tending to zero). Hence, based upon the earlier argument and this plot, we take $k$ as ten. To corroborate this choice more, we experiment with $k$ as twenty and thirty as well. As expected, and discussed in-detail later in this section, Silhouette Values for these numbers of clusters are substantially lower than those for ten clusters.

*Third*, and final, we perform experiments to identify the suitable similarity measures to build the similarity matrix, and also verify that, as recommended, the Type-3 Laplacian matrix is the best. Table 6 below gives Silhouette Values of our modified SC for all seven similarity measures and three Laplacians when clustering the earlier presented phenotypic data into 10, 20, and 30 clusters.

From this table, it is evident that Silhouette Values for the Euclidean, Squared Euclidean, City-block and Correlation similarity measures and the Type-3 Laplacian matrix are the best. Hence, we use these four similarity measures and this Laplacian matrix. Also, as mentioned earlier, Silhouette Values decrease for twenty and thirty cluster sizes.

**Table 7 Loss of accuracy because of pivotal sampling in modified SC for cluster size ten.**

| Sample size | Similarity measure | Modified SC | Modified SC with pivotal sampling | Percentage loss of accuracy (%) |
|---|---|---|---|---|
| N = 500 | Euclidean | 0.2422 | 0.2152 | −11.15 |
| | Squared Euclidean | 0.3836 | 0.3362 | −12.36 |
| | City-block | 0.2647 | 0.2369 | −10.50 |
| | Correlation | 0.3426 | 0.3367 | −1.72 |
| N = 300 | Euclidean | 0.2422 | 0.2104 | −13.13 |
| | Squared Euclidean | 0.3836 | 0.3280 | −14.49 |
| | City-block | 0.2647 | 0.2392 | −9.63 |
| | Correlation | 0.3426 | 0.3368 | −1.69 |

# RESULTS AND DISCUSSION

Using the earlier presented dataset, and sampling-clustering setups, we compare our proposed algorithm (*i.e.* modified Spectral Clustering (SC) with Pivotal Sampling) with the existing variants in four ways. Again, as earlier, we use Silhouette Values for comparison. Quantifying statistical difference between different Silhouette Values is a hard task. In general, the more closer these values are to one, the better is the clustering (see Section "Clustering Setup").

First, we demonstrate that use of sampling with modified SC does not deteriorate the quality of clustering. Second, we compare our algorithm with modified SC with Vector Quantization (VQ), Hierarchical Clustering (HC)[4] with Pivotal Sampling and HC with VQ for a sample size of 500. Since the results for modified SC with VQ come out to be closest to our algorithm, next, for broader appeal we compare these two algorithms for a sample size of 300. Third, we compare our algorithm with the current best in literature for this kind of data (*i.e.* HC without sampling) for both the sample sizes of 500 and 300. Fourth and finally, as discussed in the Literature Review section, we compare our sampling with that in *Gireesh et al. (2015)* and our clustering with the one in *Islam et al. (2020)*.

*Initially*, we calculate the loss of accuracy incurred because of Pivotal Sampling in our algorithm. This loss for both the sample sizes and cluster size ten is listed in Table 7. Columns 1 and 2 give the sample sizes and the similarity measures chosen, respectively. Columns 3 and 4 give the Silhouette Values for modified SC without sampling (from Table 6) and our algorithm, respectively. The last column gives the percentage loss of accuracy. We can observe from this data that the loss of accuracy for one type of similarity measure (Correlation) is almost as low as −2% for both the sample sizes. This is considered acceptable because we are still better than the existing best algorithm (HC without sampling; please see Table 10 and its accompanying discussion below).

Here, we also perform a statistical test to support the above conjecture that using Pivotal Sampling does not substantially deteriorate the accuracy of our modified SC. For this, we use the ANOVA (analysis of variance) test (*Rutherford, 2011*). This test uses the variance between the different groups and the variance within each group to compute a value called the F-value, which is then compared with a standard estimate called

[4] HC also requires building a similarity matrix.

**Table 8 Silhouette values for modified SC and HC with pivotal sampling and VQ for $N = 500$.** Silhouette values in bold represent good clustering. Silhouette values marked with * represent inflated values.

| Sr. No. | Similarity measure | # of clusters ($k$) | modified SC | | HC | |
|---------|--------------------|--------------------|----|----|----|----|
| | | | Pivotal sampling | VQ | Pivotal sampling | VQ |
| 1. | Euclidean | 10 | **0.2152** | 0.2061 | 0.2105 | −0.1040 |
| | | 20 | **0.1905** | 0.1448 | 0.2263* | −0.1620 |
| | | 30 | **0.1741** | 0.1021 | 0.1933* | −0.2874 |
| 2. | Squared Euclidean | 10 | **0.3362** | 0.2969 | 0.2634 | −0.2096 |
| | | 20 | **0.2469** | 0.1522 | 0.3726* | −0.5899 |
| | | 30 | **0.1658** | 0.0440 | 0.2933* | −0.6083 |
| 3. | City-block | 10 | **0.2369** | 0.2354 | 0.1703 | −0.2278 |
| | | 20 | **0.2019** | 0.1870 | 0.1879 | −0.2398 |
| | | 30 | **0.1752** | 0.1524 | 0.1988* | −0.2868 |
| 4. | Correlation | 10 | **0.3367** | 0.2560 | 0.2582 | −0.0060 |
| | | 20 | **0.2291** | 0.0899 | 0.0867 | −0.4120 |
| | | 30 | **0.1742** | −0.0349 | 0.0998 | −0.7018 |

F-critical. If F-value is less than F-critical, then it is inferred that the means of all the groups are equal.

The two groups for us refer to the modified SC results (column 3) and the modified SC with Pivotal Sampling results (column 4). The F-values here (using the Silhouette Values of the two groups) come out to be 0.3432 and 0.4202 for $N = 500$ and $N = 300$, respectively. Both these values are less than the F-critical value given in the F-distribution table of *Beyer (2019)*, which is 5.9873. Thus, using the above mentioned ANOVA test theory, we infer that that the mean Silhouette Value of modified SC is similar to the mean Silhouette Value of modified SC with Pivotal Sampling for both the sample sizes.

The results for the *second* set of comparisons are given in Table 8. Columns 2 and 3 give the similarity measures and the number of clusters chosen, respectively. Columns 4 and 5 give Silhouette Values of modified SC with Pivotal Sampling and VQ, respectively, while columns 6 and 7 give Silhouette Values of HC with Pivotal Sampling and VQ, respectively.

When we compare our algorithm (values in the fourth column, and highlighted in bold) with other variants, it is evident that we are clearly better than modified SC with VQ and HC with VQ (values in the fifth and the seventh columns); all our values are higher than those from these two algorithms.

When we compare our algorithm with HC with Pivotal Sampling (values in the sixth column), we again perform better for many cases. However, for some cases, our algorithm performs worse than HC with Pivotal Sampling (highlighted with a *). Upon further analysis (discussed below), we realize that segregation of species by HC with Pivotal Sampling into fewer clusters than practically observed, results in these set of Silhouette Values getting wrongly inflated.

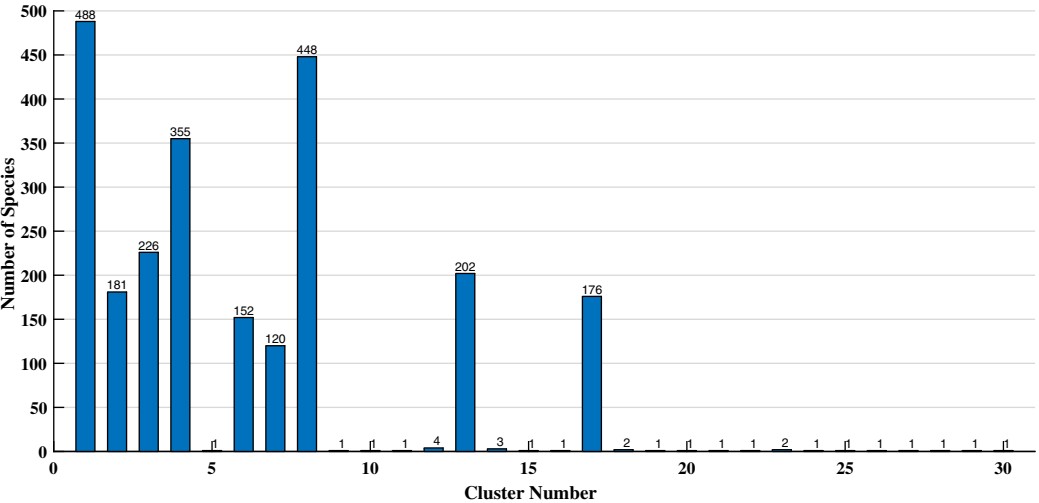

**Figure 2 Distribution of species (HC with pivotal sampling) for squared Euclidean similarity measure and cluster size thirty.**

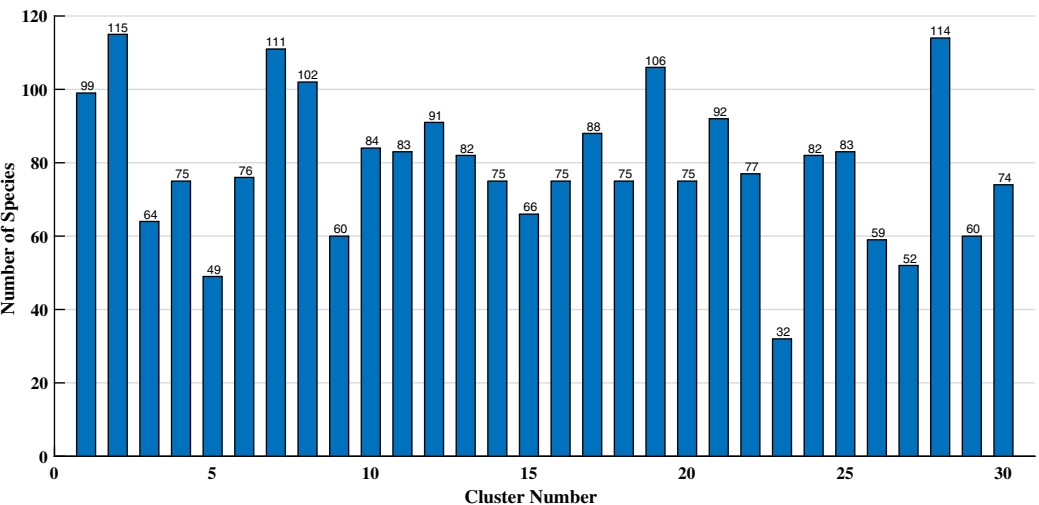

**Figure 3 Distribution of species (modified SC with pivotal sampling) for squared Euclidean similarity measure and cluster size thirty.**

To further assess the quality of the proposed technique, we present the distribution of species into different clusters (after reverse-mapping) for HC with Pivotal Sampling and our algorithm. Without loss of generality, this comparison is done using the Squared Euclidean similarity measure and cluster size thirty. The results for HC with Pivotal Sampling are given in Fig. 2 and for our algorithm are given in Fig. 3. In both the figures, on the *x*-axis, we have the cluster number and on the *y*-axis, the number of species present in them.

As evident, Fig. 2 depicts a very skewed distribution, *i.e.* most species are segregated into only a few clusters, while the remaining clusters contain only one or two species. At a broader level, this biased distribution of species obtained by HC with Pivotal Sampling is correct since all species belong to the same plant. On the contrary, the distribution in Fig. 3

**Table 9 Silhouette values for modified SC with pivotal sampling and VQ for N = 300.**

| Sr. No. | Similarity measure | # of clusters (k) | Modified SC | |
|---|---|---|---|---|
| | | | Pivotal sampling | VQ |
| 1. | Euclidean | 10 | 0.2104 | 0.1833 |
| | | 20 | 0.1968 | 0.0955 |
| | | 30 | 0.1743 | 0.0722 |
| 2. | Squared Euclidean | 10 | 0.3280 | 0.2589 |
| | | 20 | 0.2424 | 0.1322 |
| | | 30 | 0.1613 | 0.0044 |
| 3. | City-block | 10 | 0.2392 | 0.2157 |
| | | 20 | 0.1990 | 0.1696 |
| | | 30 | 0.1752 | 0.1373 |
| 4. | Correlation | 10 | 0.3368 | 0.2229 |
| | | 20 | 0.2312 | 0.0336 |
| | | 30 | 0.1725 | −0.0788 |

**Table 10 Silhouette values of modified SC with pivotal sampling and HC for cluster size ten.**

| Sample size | Similarity measure | Modified SC with pivotal sampling | HC | Percentage improvement (%) |
|---|---|---|---|---|
| N = 500 | Euclidean | 0.2152 | 0.2173 | −0.97 |
| | Squared Euclidean | 0.3362 | 0.3257 | 3.22 |
| | City-block | 0.2369 | 0.2135 | 10.96 |
| | Correlation | 0.3367 | 0.2307 | 45.95 |
| N = 300 | Euclidean | 0.2104 | 0.2173 | −3.28 |
| | Squared Euclidean | 0.3280 | 0.3257 | 0.71 |
| | City-block | 0.2392 | 0.2135 | 12.04 |
| | Correlation | 0.3368 | 0.2307 | 45.99 |

is fairly equal. That is, our algorithm equally distributes all species between the different clusters. At a finer level, this distribution is better since our algorithm is able to perform a more detailed clustering, *i.e.* it splits the bigger clusters into multiple smaller ones, which better captures the similarity between species.

This is also the reason for the inflation of Silhouette Values of HC with Pivotal Sampling in Table 8 since the intra-cluster similarity for solitary specie is zero leading to its respective Silhouette Value to become one (the maximum possible; see Eq. (21)). Thus, our algorithm also outperforms HC with Pivotal Sampling, which from Table 8 was not very evident.

Next, as mentioned earlier, to further demonstrate the applicability of our work, we also present the results with a sample size 300. Since modified SC with VQ turns out to be our closest competitor, we compare our algorithm with this one only. This comparison is given in Table 9, with its columns mapping the respective columns of Table 8. As evident

from Table 9, our modified SC with Pivotal Sampling substantially outperforms modified SC with VQ (see values in columns 4 and 5).

As earlier, *third*, we compare the results of our algorithm (modified SC with Pivotal Sampling) with the currently popular clustering algorithm in the plant studies domain (*i.e.* HC without sampling). For this set of experiments, without loss of generality, we use the cluster size of ten. The results of this comparison are given in Table 10, where the first four columns are self-explanatory (based upon the data given in Tables 8 and 9 earlier). In the last column of this table, we also evaluate the percentage improvement in our algorithm over HC. As evident from this table, our algorithm is up to 45% more accurate than HC for both the sample sizes. As earlier, our algorithm also has the crucial added benefit of reduced computational complexity as compared to HC.

*Fourth* and *finally*, as mentioned in the Literature Review section, we also compare our work with two previous works that are closest to ours. With the dataset almost the same as used by us, that is, a slightly larger phenotypic data for Soybean species, *Gireesh et al. (2015)* performed Principal Component and Power Core based samplings to identify relationships between the different phenotypic characteristics (first category as in Section "Literature Review"). We compare our sampling results with the best from *Gireesh et al. (2015)* in Appendix B, which demonstrates the superiority of our sampling method. *Islam et al. (2020)* performed HC on phenotypic data for Rice species (second category as in Section "Literature Review"). In Appendix C, we apply modified SC on this dataset to again demonstrate that our clustering technique is better.

## CONCLUSIONS AND FUTURE WORK

We present the modified Spectral Clustering (SC) with Pivotal Sampling algorithm for clustering plant species using their phenotypic data. We use SC for its accurate clustering and Pivotal Sampling for its effective sample selection that in-turn makes our algorithm scalable for large data. Since building the similarity matrix is crucial for the SC algorithm, we exhaustively adapt seven similarity measures to build such a matrix. We also present a novel way of assigning probabilities to different species for Pivotal Sampling.

We perform four sets of experiments on about 2,400 Soybean species that demonstrate the superiority of our algorithm. *First*, we compare the Silhouette Values of modified SC without and with Pivotal Sampling, and show that the difference between these values is not significant. *Second*, when compared with the competitive clustering algorithms with samplings (SC with Vector Quantization (VQ), Hierarchical Clustering (HC) with Pivotal Sampling, and HC with VQ), Silhouette Values obtained when using our algorithm are higher. *Third*, our algorithm doubly outperforms the standard HC algorithm in terms of clustering accuracy and computational complexity. We are up to 45% more accurate and an order of magnitude faster than HC. *Fourth* and *finally*, we illustrate the excellence of our algorithm by comparing it with two previous works that are closest to ours.

Since the choice of the similarity matrix has a significant impact on the quality of clusters, in the future, we intend to adapt other ways of constructing this matrix such as Pearson $\chi^2$, Squared $\chi^2$, Bhattacharyya, Kullback-Liebler etc. (*Cha, 2007*). Furthermore, we

also plan to observe the performance of Cube Sampling, which is another probabilistic sampling technique with data analysis properties complementary to Pivotal Sampling (*Tille, 2006*). Both Pivotal and Cube belong to the balanced sampling category, *i.e.* they satisfy $Y \approx Y'_{HT}$ and $Y \approx Y'_{H\acute{a}jek}$ (recall Eqs. (18)–(20)). Cube Sampling automatically obtains the samples (without specifying the sample size), which does not happen in Pivotal. As mentioned earlier, our algorithm is developed to work well for phenotypic data of all plant species. This is because different species vary only in the number of characteristics and the type of characteristics, both of which do not affect our algorithm. We have preliminarily discussed this aspect for Maize and Rice in Appendix C, with extensive experiments for these two plants planned for future.

## APPENDIX A

Here, we first present phenotypic data of the Soybean species used for our experiments. Please see Table A1 below. Next, we validate this data. For this, we compare our species data with a similar Soybean species data from (*Gireesh et al., 2015*) for the common set of phenotypic characteristics; Plant Height (PH), Number of Pods Per Plant (NPPP), and Days to Pod Initiation (DPI). This comparison is done using standard statistical metrics and is given in Table A2 below.

From this table, it is evident that the Standard Deviation (SD), Coefficient of Variance (CV), and Mean of our data and the data from the previous work are very close (for all three characteristics of PH, NPPP, and DPI). The slight variation in the metrics between the two data for all the characteristics is due to the difference in the ranges of the respective characteristics (due to the slightly differing selection of the species by the two works).

## APPENDIX B

Here, we compare our sampling technique with those proposed by *Gireesh et al. (2015)* for a similar dataset. As earlier, we do Pivotal Sampling on 2,376 Soybean species while *Gireesh et al. (2015)* performed the Principal Component Score (PCS) and the Power Core (PC) samplings on 3,443 Soybean species. Since the samples obtained by the PC method are better, we compare our results with this sampling only.

This comparison is done using the statistical metrics of Standard Deviation (SD), Coefficient of Variance (CV) and Mean, and is given in Table A3 below. Since the metrics of our sampled data are more closer to our respective full data as compared to the metrics of the previous works' sampled data to its respective full data, our sampling is better.

## APPENDIX C

In the article, we have demonstrated the usefulness of our algorithm on the species of the Soybean plant, here we demonstrate our algorithms' applicability to the species of the other two plants (Maize and Rice). The phenotypic data for the Maize species is given in Table A4, and for the Rice species is given in Table A5.

We can observe from Tables A1, A4, and A5 that there is a set of common phenotypic characteristics for the three plant species. Also, the values of all the characteristics are

**Table A1  Phenotypic data of the Soybean species used for experiments.** EPV, Early Plant Vigor; PH, Plant Height; NPB, Number of Primary Branches; LS, Lodging Score; NPPP, Number of Pods Per Plant; SW, 100 Seed Weight; SYPP, Seed Yield Per Plant; DPI, Days to Pod Initiation.

| Species | EPV | PH | NPB | LS | NPPP | SW | SYPP | DPI |
|---------|-----|-----|-----|-----|------|-----|------|-----|
| 1 | Poor | 54 | 6.8 | Moderate | 59.8 | 6.5 | 2.5 | 65 |
| 2 | Poor | 67 | 3.4 | Severe | 33 | 6.2 | 3.9 | 64 |
| 3 | Good | 60.8 | 4 | Moderate | 34.6 | 6.1 | 3 | 65 |
| ⋮ | ⋮ | ⋮ | ⋮ | ⋮ | ⋮ | ⋮ | ⋮ | ⋮ |
| $n$ | Very Good | 89.6 | 5 | Severe | 32.6 | 7.3 | 3.4 | 62 |

**Table A2  Comparison of SD, CV, mean, and range for our phenotypic data and similar previous data.** Here, for comparison purposes, we have to work with original (non-normalized) values of the characteristics.

| Parameter | Work | PH | NPPP | DPI |
|-----------|------|-----|------|-----|
| Standard Deviation (SD) | Our Work | 16.61 | 20.16 | 7.85 |
| | Previous Work (*Gireesh et al., 2015*) | 18.6 | 24.1 | 8 |
| Coefficient of Variance (CV) | Our Work | 31.80 | 47.13 | 13.62 |
| | Previous Work (*Gireesh et al., 2015*) | 30.9 | 55.2 | 17.8 |
| Mean | Our Work | 52.24 | 42.78 | 57.60 |
| | Previous Work (*Gireesh et al., 2015*) | 60.3 | 43.6 | 54.7 |
| Range | Our Work | 13–102 | 4.33–197.66 | 24–80 |
| | Previous Work (*Gireesh et al., 2015*) | 5.4–118.8 | 1.33–301 | 30–98 |

**Table A3  Comparison of pivotal sampling and power core method for three characteristics.** Here, for comparison purposes, we have to work with original (non-normalized) values of the characteristics.

| Parameters | Work | Population | PH | NPPP | DPI |
|------------|------|-----------|-----|------|-----|
| Standard Deviation (SD) | Our Work | Overall | 16.61 | 20.16 | 7.85 |
| | | Sampled | 17.34 | 18.90 | 7.42 |
| | Previous Work (*Gireesh et al., 2015*) | Overall | 18.6 | 24.1 | 8 |
| | | Sampled | 22.15 | 45.33 | 11.73 |
| Coefficient of Variance (CV) | Our Work | Overall | 31.80 | 47.13 | 13.62 |
| | | Sampled | 31.91 | 43.97 | 13.03 |
| | Previous Work (*Gireesh et al., 2015*) | Overall | 30.9 | 55.2 | 17.8 |
| | | Sampled | 39.86 | 91.06 | 25.46 |
| Mean | Our Work | Overall | 52.24 | 42.78 | 57.60 |
| | | Sampled | 54.34 | 42.99 | 56.94 |
| | Previous Work (*Gireesh et al., 2015*) | Overall | 60.3 | 43.6 | 54.7 |
| | | Sampled | 55.57 | 49.78 | 56.65 |

**Table A4 Phenotypic data of the Maize species (*Belalia et al., 2019*).** DS, Days to Silking; PH, Plant Height; EH, Ear Height; ED, Ear Diameter; EL, Ear Length; SW, 100 Seed Weight.

| Species | DS | PH | EH | ED | EL | SW |
|---|---|---|---|---|---|---|
| 1 | 77 | 75 | 33 | 3.2 | 11.6 | 2.3 |
| 2 | 98 | 45 | 14 | 2.7 | 8.1 | 1.6 |
| 3 | 68 | 132 | 80 | 3.7 | 16.2 | 3.6 |
| ⋮ | ⋮ | ⋮ | ⋮ | ⋮ | ⋮ | ⋮ |
| $n$ | 70 | 50 | 35 | 3.1 | 10.6 | 2.6 |

**Table A5 Phenotypic data of the rice species (*Islam et al., 2020*; *Kim, 2019*).** TN, Tiller Number; PH, Plant Height; PN, Panicle Number; PL, Panicle Length; SW, 100 Seed Weight; BDR, Blast Disease Resistance.

| Species | TN | PH | PN | PL | SW | BDR |
|---|---|---|---|---|---|---|
| 1 | 6.8 | 124.2 | 5.5 | 25.6 | 22.1 | Resistant |
| 2 | 6.5 | 121.6 | 6.8 | 24.8 | 23.1 | Moderately Resistant |
| 3 | 7.2 | 126.4 | 4.5 | 26.1 | 19.5 | Moderately Susceptible |
| ⋮ | ⋮ | ⋮ | ⋮ | ⋮ | ⋮ | ⋮ |
| $n$ | 7.1 | 131.4 | 5.1 | 25.9 | 18.5 | Susceptible |

**Table A6 Silhouette values of modified SC and HC for three clusters of ten rice species.**

| Similarity Measure | modified SC | HC |
|---|---|---|
| Euclidean | 0.2743 | 0.0076 |
| Squared Euclidean | 0.3276 | 0.0253 |
| City-block | 0.2561 | 0.0219 |
| Correlation | 0.3265 | 0.0433 |

either categorical or numerical. As mentioned earlier, the categorical values can be easily converted to numerical ones. Since the input to our algorithm is a matrix built using the phenotypic data for given species, it can be applied to any of these plants.

To demonstrate the usefulness of our algorithm to the two new plant species, without loss of generality, we perform clustering of Rice species using our modified SC. For this, we use the data from *Islam et al. (2020)*, where the authors have used HC to cluster ten Rice species into three clusters. Hence, we also cluster these ten species into three clusters using our modified SC. In *Islam et al. (2020)*, the output is in the form of a hierarchical tree, which is non-numerical, and hence, difficult to compare. Thus, we compute Silhouette Values for our modified SC and HC. This data for the four similarity measures are given in Table A6. As evident from this table, our algorithm substantially outperforms HC.

## ACKNOWLEDGEMENTS

The authors would like to thank Mr. Mohit Mohata, Mr. Ankit Gaur and Mr. Suryaveer Singh (IIT Indore, India) for their help in preliminary experiments, which they did as part of their undergraduate degree project. We would also like to sincerely thank Dr. Vangala Rajesh and Dr. Sanjay Gupta (Indian Institute of Soybean Research, Indore, India) for their help in generating the experimental data.

### Funding

This work was supported by the Ministry of Electronics and Information Technology (MeitY), India under the Visvesvaraya PhD Scheme for Electronics & IT and the MATRICS Scheme of Department of Science and Technology (DST-SERB), India with project number MTR/2017/001023. The funders had no role in study design, data collection and analysis, decision to publish, or preparation of the manuscript.

### Grant Disclosures

The following grant information was disclosed by the authors:
Ministry of Electronics and Information Technology (MeitY), India.
MATRICS Scheme of Department of Science and Technology (DST-SERB), India: MTR/2017/001023.

### Competing Interests

The authors declare that they have no competing interests.

### Author Contributions

- Aditya A. Shastri conceived and designed the experiments, performed the experiments, analyzed the data, prepared figures and/or tables, authored or reviewed drafts of the paper, and approved the final draft.
- Kapil Ahuja conceived and designed the experiments, performed the experiments, analyzed the data, prepared figures and/or tables, authored or reviewed drafts of the paper, and approved the final draft.
- Milind B. Ratnaparkhe analyzed the data, authored or reviewed drafts of the paper, and approved the final draft.
- Yann Busnel conceived and designed the experiments, authored or reviewed drafts of the paper, and approved the final draft.

### Data Availability

The phenotypic data used for experiments, in human-readable form, and the MATLAB scripts for all the experiments, with data in computer-readable form, are available in the Supplemental Files.

## Supplemental Information

Supplemental information for this article can be found online at http://dx.doi.org/10.7717/peerj.11927#supplemental-information.

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
