# Peer review of "Probabilistically sampled and spectrally clustered plant species using phenotypic characteristics"

_PeerJ, doi:10.7717/peerj.11927_

## Round 0.1 · original submission · Major Revisions

Thank you for submitting this paper. The reviewers believe it has scientific merit but are in general agreement that you need substantial edits to refine it. Please work carefully through the suggestions made in the reviews and ensure you address them fully.

Reviewer 1 ·

Basic reporting

There contains grammatical errors and typos in the manuscript. The authors should re-check and revise carefully.

The authors listed a lot of references in the literature review. However, in my opinion, they are listed one by one. It is better if the authors could group the literature review according to the methods or data.

Experimental design

Equations should be assigned to their numbers.

There are a lot of paragraphs that should be in the "Methodology" section, but they are in "Results" now. For example, "Sampling Discussion" part, "Clustering setup", etc.

Validity of the findings

Table 1 did not contain any meaning if you only listed a few phenotypes. It is necessary to show all phenotypes, maybe in supplementary.

How did the authors convince that their methods could be applied to any phenotype? It needs some evidence.

Is there any statistical tests had been performed for comparison among different methods?

The authors should compare to previously published works/methods on the same dataset.

The authors should have some validation data.

Additional comments

No comment

Reviewer 2 ·

Basic reporting

The author’s use of English was generally competent, although the sentence structure could be tightened. Specific instances where I disagree with author word usages are detailed in the general comments below. Background intended to show the context of this work in relation to plant breeding lacks relevant detail, and so is sometimes misleading. The literature review is repetitive and relevant literature could be grouped where similar.

Experimental design

The research question was clearly stated. The experimental design appears appropriate to the objectives. However, the set of analyses performed and what hypothesis each was intended to test is not clearly presented.

Validity of the findings

The authors should be more candid when comparing their methods to other methods. There is no way of knowing, for example, which silhouette values are statistically different from one-another. Is the contrast between silhouette values .2152 and .2105 (Euclidean distance with k=10) really different compared with the contrast of .3367 and.2582 (correlation with k=10)? Data comparing computational complexity of the various methods is not presented, yet the authors conclude that their method is faster. They should present some data.

Additional comments

line 23: Genome data should more properly be called genotypic data – referring to sets of genotype calls. The authors have a nomenclature problem. By referring to individual soybean pure lines as genotypes, they were forced to use the incorrect ‘genome data’ term. Genotype is often used as a generic term for pure lines (or clonal lines or hybrid lines), but the two are confusing together. I suggest that they either specifically define the meaning of the term plant genotype in the context of this paper or use a different word to describe an individual or leave genotypic data out of the introduction altogether.
lines 35-37: The relationship between breeding progress and genetic diversity is much more complicated than stated. For a more nuanced review I suggest (among others): Louwaars, NP. 2018. Plant breeding and diversity: A troubled relationship? Euphytica 214,114. For a better defense of the role of genetic diversity in plant breeding: Swarup, S., Cargill, E.J., Crosby, K., Flagel, L., Kniskern, J., and Glenn, K.C. 2020. Genetic diversity is indispensable for plant breeding to improve crops. Crop Science. https://doi.org/10.1002/csc2.20377
lines 38-40: Incomplete. I presume the authors are referring to genotype-by-sequencing methods of acquiring genotypic data on hundreds of thousands of SNPs for hundreds of samples. This is something different from genome sequencing. They should use more specific language.
line 41: I suggest you say “Plant genetic diversity can also be studied…”
line 51: I wasn’t sure what you meant by “genotypes of the same plant”, please clarify.
lines 91-92: Please avoid phrasing correlation in a way that implies causation – your example comes too close to this. Also “lower” not “less” here and throughout.
lines 112-114: when you say “activity on” do you mean “correlation with”? If not, then more detail on how the authors proved causation should be included.
lines 118-119: I don’t know what you are trying to say with this sentence. Wheat breeders are people, therefore diverse rather than related (except in a few well-noted cases). Were you trying to talk about genetic distance within and between wheat breeding programs?
line 137: Not “catered”, perhaps “addressed”?
line 250-251: should be “…since the better the matrix quality, the better the clustering accuracy”.
Line 315: Did you convert non-numerical values into numerical values or did you remove traits with non-numerical values?
Line 360: “lower” not “lesser”
Table 4 legend: define the meaning of * here as well as in the text.
Line 433: Sine you state above that your method worked well, why do you propose replacing Pivotal Sampling?

·

Basic reporting

The introduction was succinct and the related literature well-reviewed.
Based on the methodology of this study, it will be interesting to see the result of spectral clustering on the complete samples in the population without prior application of sampling. If we have a less significant difference in the results this will reduce the time and design complexity of the experiments.

Deviation property of the genes was used as the inclusion criterion of the pivotal sampling, can you specify the threshold chosen for the selection and rejection step in line 162-169?

Experimental design

I think this study is purely computational and not require experimental design.

Validity of the findings

Line 395-400: Further information from the validation done will be required to justify the conclusion about the skewness of the clustering distribution not being a true representation of the samples. The segregation of the samples in a few clusters might be expected since all genotype belongs to the same plant. Kindly include these details in the manuscript.

In addition to the normalization performed in the data preprocessing phase, the one-hot encoding of the categorical variables (EPV and LS) can be performed in future studies since the number of attributes are few.

Additional comments

In general, the manuscript is well written.

---

## Round 0.2 · accepted · Accept

The original Academic Editor is not available and so I have stepped in to handle the final step in the review process.

Your paper has now been reviewed again by the two previous reviewers. I am happy to report that they are both satisfied with your revision and feel you have addressed the previous concerns. I am, therefore, happy to now recommend acceptance of your paper. Congratulations.

Reviewer 1 ·

Basic reporting

No comment.

Experimental design

No comment.

Validity of the findings

No comment.

Additional comments

My previous comments have been addressed.

·

Basic reporting

No comment

Experimental design

No comment

Validity of the findings

No comment

Additional comments

I have gone through the revised manuscript, all the issues raised have been attended to by the authors.
I am just suggesting in the "Results and Discussion" section if the "Discussion" part can have a separate section would be fine.

Aside from this, it is highly recommended for publication.